# The Physiological Implications of *S*-Nitrosoglutathione Reductase (GSNOR) Activity Mediating NO Signalling in Plant Root Structures

**DOI:** 10.3390/antiox9121206

**Published:** 2020-11-30

**Authors:** Leslie Ventimiglia, Bulent Mutus

**Affiliations:** Department of Chemistry and Biochemistry, University of Windsor, 401 Sunset Ave., Windsor, ON N9B 3P4, Canada; ventimi@uwindsor.ca

**Keywords:** GSNOR, nitrate, root physiology, auxin, phytohormones

## Abstract

Nitrogen remains an important macronutrient in plant root growth due to its application in amino acid production, in addition to its more elusive role in cellular signalling through nitric oxide (NO). NO is widely accepted as an important signalling oxidative radical across all organisms, leading to its study in a wide range of biological pathways. Along with its more stable NO donor, *S*-nitrosoglutathione (GSNO), formed by NO non-enzymatically in the presence of glutathione (GSH), NO is a redox-active molecule capable of mediating target protein cysteine thiols through the post translational modification, *S*-nitrosation. *S*-nitrosoglutathione reductase (GSNOR) thereby acts as a mediator to pathways regulated by NO due to its activity in the irreversible reduction of GSNO to oxidized glutathione (GSSG) and ammonia. GSNOR is thought to be pleiotropic and often acts by mediating the cellular environment in response to stress conditions. Under optimal conditions its activity leads to growth by transcriptional upregulation of the nitrate transporter, NRT2.1, and through its interaction with phytohormones like auxin and strigolactones associated with root development. However, in response to highly nitrosative and oxidative conditions its activity is often downregulated, possibly through an *S*-nitrosation site on GSNOR at cys271, Though GSNOR knockout mutated plants often display a stunted growth phenotype in all structures, they also tend to exhibit a pre-induced protective effect against oxidative stressors, as well as an improved immune response associated with NO accumulation in roots.

## 1. Introduction

As immobile organisms, plants must endure the brunt of the environment to which they are exposed. Unable to evade harsh conditions, they have developed pathways with which to respond to environmental stressors, allowing them to maintain homeostasis even under unfavorable conditions. Plants are finely tuned products of evolution, able to respond to many biological stressors at once, such as changes in temperature, soil nutrition, acidity, salinity, metals, pathogens, and physical damage. Responses to stressors often result in a physical change in the plant itself to mitigate harmful effects, often causing stunted growth in favor of survival [1]. In response to the many ways they may endure stress, plants often employ reactive oxygen species (ROS), and reactive nitrogen species (RNS), to mediate stress response pathways [2]. At the center of ROS and RNS response pathways is the enzyme *S*-nitrosoglutathione reductase (GSNOR), a primary regulator of *S*-nitrosoglutathione (GSNO), the principle substrate for GSNOR. By regulating GSNO levels, GSNOR is capable of mediating the amount of *S*-nitrosothiols (SNOs) presenting on cysteine and methionine thiols, a common reversible post-translational modification (PTM) capable of altering enzyme activity and possibly regulating many of the responses to stressors described previously [3].

The formation of GSNO is a favorable reaction occurring non-enzymatically in oxidizing environments where there is a high concentration of nitric oxide (NO) and glutathione (GSH), the principle tripeptide antioxidant in eukaryotic cells [3,4]. NO has been identified as a vital regulator in many biological processes. It is a highly reactive free radical which can cross lipid bilayers and react directly with the thiols of cysteine groups to form SNOs.NO has also been identified to interact with the tyrosine aromatic ring during times of high oxidative stress [2]. SNOs have been recently identified as an important mechanism by which enzymes are regulated in response to stress [2]. Because NO degrades quickly, GSNO, a much more stable molecule, is often considered a low molecular weight reservoir of bioactive NO essential to NO-dependent signalling which is capable of transporting NO through the vasculature of the plant [2,5], and transferring its nitroso group onto thiols similarly to free NO through the process of trans-nitrosation. However NO and GSNO do not always interact with the same protein thiols, indicating a possible regulation mechanism where GSNO concentrations may act as a regulatory mechanism for maintaining proper levels of protein *S*-nitrosation [2,3]. Because GSNOR is capable of irreversibly lowering GSNO levels, it is thought to indirectly control the concentration of SNOs present, making it an important regulatory enzyme in stress response [2,3].

GSNOR is present in the cytosol and a highly conserved enzyme amongst eukaryotes and prokaryotes. It is capable of the NADH-dependent reduction of GSNO to glutathione disulphide (GSSG), the oxidized form of GSH, and ammonium (NH_3_) [3]. It was originally identified in plants as a glutathione-dependent formaldehyde dehydrogenase (FALDH), and a member of the class III alcohol dehydrogenase family, where the primary substrate is hemithioacetal *S*-hydroxymethylglutathione (HMGSH), which is formed in an oxidizing environment through the favorable reaction of formaldehyde and GSH, using a catalytic zinc, and in the presence of the coenzyme NAD^+^ [3,6]. The redox-active enzyme acts in the homeostasis of *S*-nitrosothiols (SNOs) and is capable of regulating many cellular processes in that manner.

## 2. Root Development in Response to Changes in Nutrient Availability

Plant development proceeds in conjunction with its immediate environment. Under the soil surface, root systems are directly in contact with available micro and macro nutrients as well as possible pathogens, toxins and symbiotic organisms present in the rhizosphere, the physical area immediately surrounding the root structure [7]. The rhizosphere has the potential to change the architecture of root systems, which ultimately alters a plants ability to take up nutrients and water, as well as interact with possible stressors. Root structures are not predetermined genetically, they may be altered through cell division at the primary root (PR), lateral roots (LR), and modifications to the numbers of root hairs presenting on root surfaces to adapt to the levels of nutrients available in its environment [7,8,9], a factor of plant environmental adaptation referred to as its plasticity.

Nitrogen, a plant macronutrient in the soil, is also thought to act as a limiting factor in plant growth due to its significant role in the biosynthesis of amino acids for cellular growth. Low nitrogen soil can often lead to stunted growth and low plant yield. It often must be supplemented into the soil in the form of fertilizer. Limiting factors, like nitrogen, phosphorus, sulfur and other necessary elements to plant growth are thought to alter root architecture to adapt to these changes in nutrient availability [7]. Conversely, high levels of nitrogen availability are linked to increases in pathogen interactions at the root level due to their similar nutrient requirements [10], leading to a possible immune response pathway in root structures to mitigate pathogen effects at high nitrogen levels [11]. Though some studies have determined that GSNOR-inhibited *Arabidopsis* plants do not tend to show reductions in root growth [11], others suggest that GSNOR knock out plants undergo a plethora of phenotypic changes, including root growth inhibition, reduction in photosynthesis and decreased fruit productivity [12]. Despite phenotypic changes, at increased soil-nitrogen content, GSNOR knock out plants are linked to an immune response, associated with an accumulation of NO, GSNO and lower salicylic acid (SA) thereby improving disease resistance [11,13].

Additionally, structural changes have been observed under iron deficient conditions, where GSNOR is thought to be upregulated [14]. Under these conditions NO, GSNO, and GSH decrease, leading to changes in growth probably regulated by GSNOR localized in the phloem [14]. However, GSNOR is also required in times of iron toxicity, where GSNOR knock out strains displayed reduced primary root growth under high iron conditions but relatively no change in wild type (WT) seedlings [15].

Nitrogen is thought to enter plant systems through their roots in the form of nitrate (NO^3-^), where in low nitrate conditions, lateral and primary root elongation has been observed to be higher than in a high nitrate medium [7,16]. These results (Figure 1) suggest a response mechanism between nitrate uptake and plant growth hormone levels, such as levels in the phytohormones auxin and cytokinin, which have been associated with development concentrated at the root and shoot structures respectively [17]. Studies have indicated a possible relationship between nitrate levels and auxin response pathways. In experimentation, *Arabidopsis* auxin resistant mutant axr4 roots were reported to be desensitized to the inhibitory effect of high nitrate levels on the WT lateral roots [18], however some studies suggest the opposite effect [16].

Due to the effects of nitrates on root architecture, it is suggested that nitrate is linked to a signalling molecule as well as a nutrient in plant growth [18]. There is ample evidence that once nitrates enter root structures through the nitrate transporter proteins NRT1.1 and NRT2.1, they can be further reduced to nitrites through nitrate reductase (NR) and even further reduced to the signalling molecule NO [19]. Though the role of GSNOR in root development is unclear, many studies suggest that NO, a precursor to its principle substrate GSNO, can mediate the eventual morphology of root structures through its relationship to the auxin signalling pathway [20] and through its connection with RNS and ROS. Though it is expressed within many plant tissues, GSNOR is thought to be localized in the phloem, and xylem parenchyma cells of the vasculature, capable of regulating NO levels throughout the plant [21].

## 3. The Role of Nitric Oxide in Plant Biology

Plants tend to be highly responsive to their nitrogen levels, a process regulated by NR [19], the first enzyme of nitrogen assimilation, as well as the principle contributor to NO availability through increasing cellular levels of nitrite (NO^2−^) [7,22]. For these reasons, nitrate reductase has been identified as a highly regulated and short-lived enzyme. Its synthesis has been associated with positive growth conditions, like light and glucose availability, whereas its downregulation can be associated with dark conditions and a surplus of amino acids [23]. Due to its biological importance as a signalling molecule, NO regulation and production has garnered a lot of research interest.

There are multiple intracellular pathways identified to contribute to NO levels. In most eukaryotes and prokaryotes, NO bioavailability is produced through constitutive and inducible forms of nitric oxide synthase (NOS), a complex enzyme capable of generating NO and citrulline from L-arginine in an oxidative environment and dependent on NADPH [3,5,24]. Conversely, NO is formed in plants mainly through nitrate (NO^3−^) reduction [5], including nitrite reductase (NiR) activity, and possible nitrite reduction identified in the chloroplast and mitochondrial matrix [22,25]. There is also evidence that NR itself could be capable of reducing nitrate to NO in land plants through electron transfer from NADPH to nitric oxide-forming nitrite reductase (NOFNiR) [26]. Through this mechanism NR has also been described to act as a potential NO scavenger by donating electrons from NADPH onto a truncated hemoglobin THB1 which can function as a dioxygenase [22].

In mammals, nitric oxide synthase is widely accepted as the principal contributor to intracellular levels of NO [27], a recently discovered nitric oxide synthase-like enzyme, is proposed as another possible source of NO in land plants [28]. The enzyme, discovered originally in green algae, has been reported to have a similarly structured active site containing L-arginine, and NADPH binding sites, along with cofactor binding sites for FAD, FMN, BH4, and CaM, in addition to a 43 percent sequence homology to human eNOS [28]. Additionally, many plant extracts have been found to contain NOS inhibitors offering further evidence in favor of an NOS-like enzyme present in land plants [27], however more recent genomic evidence does not support the activity of NOS-like enzymes as the primary contributor to intracellular levels of NO since it has only been identified across a limited number of algal genomes [27].

Nitric oxide is capable of mediating pathways by the S-nitrosation of cysteine residues of key enzymes in transcription when it increases in concentration. Additionally it is thought to be capable of contributing to signal transduction through secondary messengers like peroxynitrate (ONOO^−^) by nitrating aromatic amino acids, like tyrosine and tryptophan, during oxidative conditions [29].

## 4. GSNOR Activity is Related to NO Production

It is suggested that NO production and nitrogen assimilation pathways are interconnected through a signalling pathway Figure 2 that uses NO and possibly *S*-nitrosation of thiols as a mediator for NR as well as the nitrate transporter NRT2.1 activity [30].

Previously it has been suggested that a product of nitrate reduction, possibly NO or GSNO, regulates nitrate uptake and reduction through negative feedback [31]. Because NR is thought to be one of the main sources of NO, regulating NR activity can be associated both with nitrogen assimilation and NO generation [30]. In experimentation NR and transporter NRT2.1 activity has been shown to act in tandem with the enzyme GSNOR Figure 3 due to its ability to alter intracellular levels of GSNO [30,31].

Though NO and GSNO can exhibit similar PTMs where NO is capable of *S*-nitrosation of thiols on similar sites to GSNO, the mechanism by which this action occurs is slightly different. Where NO *S*-nitrosates proteins through a radical-mediated pathway, GSNO transnitrosates cysteine residues [32,33].

GSNO has been determined to reduce the activity of nitrate reductase by reducing nitrate levels through the downregulation of the transporter NRT2.1 in vivo [30]. In a study where NRT2.1 transcription was determined in WT, NO over producing mutant (nox1) and GSNOR1 mutant (higher GSNO levels), as well as grown in the presence of heightened GSNO level and an NO donor (DEA/NO), it was determined that while both high NO and GSNO lowered NRT2.1 transcription, GSNO caused a more significant reduction in transcription [30].

NRT2.1 is part of a group of transporters in root structures called the high-affinity transport system (HATS) [34]. Recent studies have determined that NRT2.1 and other HATS proteins are regulated through transcriptional control rather than through PTMs, however the precise mechanism for this control remains elusive [34].

Additionally, GSNOR1 in *Arabidopsis thaliana* has become a site for potential regulation through *S*-nitrosation by NO, it has been determined that high levels of NR activity tends to suppress the activity of GSNOR1 [30]. Though this process has been previously determined [35], studies by Frungillo et al. further contribute to evidence of NOs capability of modifying GSNOR1 through *S*-nitrosation.

In experiments, GSNOR1 activity is showed to decrease in response to NO donors, and biotin switch assays show evidence in favor of *S*-nitrosylation [30]. The evidence in favor of the *S*-nitrosation PTM is further supported by identification of conserved cysteine residues and mass spectral analysis of conserved sites for *S*-nitrosation [6]. In studies on *Solanum lycopersicum* conserved cys residues were identified at cys10, cys271, and cys370 Figure 4 where further mass spectrometry identified *S*-nitrosation at cys271 [36,37]. The action of increasing NO levels and GSNOR1 inhibition is often coupled with increased ROSs associated with plant immune response [35].

## 5. Strong Oxidizing Agents are Capable of Reducing GSNOR Activity

Oxidative environments often cause damage on biological systems through the oxidation of thiols and the formation of disulfide bridges, leading to hindered enzyme activity. Experimentation has determined that plant systems reversibly inhibit their GSNOR activity in response to oxidative radicals [33]. These results were apparent through comparisons between the GSNOR mutant in *Arabidopsis* and the WT GSNOR, where the GSNOR knock out was visibly capable of surviving in higher levels of oxidative herbicide paraquat (Figure 5) [38].

It is proposed that in the absence of GSNOR activity GSH production increases and its oxidation to GSSG decreases leading to higher antioxidant protection against oxidative species [38]. Additionally, the GSNOR mutant was profiled for difference in gene transcription in the GSNOR mutant versus the WT. Results show increases in peroxidases, Glutathione *S*-transferases and thioredoxin genes, and decreases in oxidoreductase transcription suggesting that the GSNOR knock out contains a pre-induced antioxidant protection system [33].

Studies have determined that the manner by which RNS are accumulated in root structures is relevant to root development and overall plant growth [39]. When NO is applied exogenously in wildtype GSNOR plants, it has been shown to promote growth in lateral root and root hair formation by selectively upregulating pathways in primordial root tissue [39]; some studies have also reported a reduction in primary root formation [20]. Conversely, the GSNOR knock out plants have exhibited overall reductions in growth where root development is thought to be directly linked to redox activity [40]. Perhaps consistent high levels of RNS within all cells GSNOR knock out plants have deleterious effects causing stunted growth [39].

## 6. Auxin is an Important Hormone Capable of Mediating Cellular Growth in Concert with GSNOR

Phytohormones are a class of light active hormones that exist at low concentration in vivo and act as plant growth regulators. Because they are sensitive to light, they allow plants to respond to light and dark conditions differently, in a similar manner to the mammalian circadian clock [41]. There are nine classes of phytohormones, including salicylic acid (SA), jasmonic acid (JA), auxin (IAA), cytokinin (CK), gibberellins (GA), abscisic acid (ABA), ethylene, brassinosteroids, and strigolactones, all known to act in tandem with NO signalling, many of which have been implicated in GSNOR regulatory pathways to control stress response throughout all aspects of plant development [42]. Of the phytohormones, auxin and cytokinin are associated with normal structural development in the roots and shoots, respectively, and act competitively to promote elongation at their respective site, a well-known and extensively studied model of plant growth [17]. Auxin signalling is specifically relevant when considering the growth of root structures in response to GSNO levels, where a mechanism has been identified to regulate TIR1 (Figure 6A), a nuclear F-box protein and the auxin receptor [1]. The auxin signalling pathway has been thought to contain a complex of TIR1, which acts as the auxin receptor, bound to AUX/IAA repressor and an auxin response factor (ARF) [1]. When auxin is bound to TIR1, the complex dissociates causing the ubiquitination of the repressor, and ARF is liberated and capable of binding to promoter regions upregulating transcription of protein targets [1,43]. The study done by Shi et al. explores a concept cited in literature that auxin signalling is increased on the TIR1 auxin receptor through an *S*-nitrosation site [44]. At increased GSNO levels, and thereby reduced GSNOR activity, *S*-nitrosation of TIR1 receptor is thought to increase its affinity for auxin and in turn increase transcription of target proteins. Through genomic and mass spectroscopic analysis of the TIR1 receptor, Terrile et al. offer support in favor of proposed *S*-nitrosation on cysteine residue cys140, however there are conflicting results displayed by Shi et al. in vivo experimentation of the GSNOR 1–3 mutant, where reduced GSNOR activity and thereby increased levels of GSNO conversely show reduced apical and lateral root growth. The juxtaposition of these results favors GSNORs identity as a pleiotropic enzyme, having effects on a multitude of pathways, and highlights the importance of finding ways to isolate pathways for study.

Auxin can also mediate growth through the manner by which it is transported through the root structures [1]. PIN proteins allow for the influx and efflux of auxin through the cells. In the roots, they cause the growth of the roots to flow in the direction of gravity [45,46]. NO is thought to act downstream to auxin in response pathways, where its production is upregulated in response to auxin. However, PIN1 auxin transport, responsible for apical growth in roots, is thought to be inhibited by NO accumulation [20]. In the study by Shi et al., the GSNOR 1–3 mutant is associated with reductions of PIN proteins leading to less auxin availability in the roots, consistent with Figure 6 showing no lateral root formation in the mutant. This is further evidence in favor of the pleiotropic effects of GSNOR [1].

## 7. Interplay between Strigolactones and NO

Similarly, to NO and the phytohormones, strigolactones (SLs) have been identified to play a role in the regulation of cellular signals during plant development and stress response. SLs are derived from carotenoid hormones, a class of C40 terpenoid pigments that enable photoactive processes and act in growth regulation [47].

Emerging studies have suggested a possible relationship between the oxidative radical, NO, and SL plant hormones [48,49]. The study of the relationship of NO and SLs is a new and evolving field of research. There are currently at least 20 known SLs which have been identified, all of which have been associated with root and shoot system development. NO has been identified as a possible negative regulator of SL production, yet thought to also act as an apparent inducer of SL signaling [48]. In a recent study of the relationship between NO, auxin, and SLs; WT and mutants for certain SLs biosynthesis pathways (max1-1 and max2-1) were exposed to a synthetic racemic analog, GR24, and measured for NO level. These results indicate that SLs might be capable of employing ROS as a method of regulation [48]. Another recent study by Oláh et al. determined that max1-1 and max2-1 resulted in increases in NO and SNO levels even without the addition of GR24 which were also associated with decreased GSNOR activity or abundance. It is thought that GSNOR may play a role in controlling SL induced primary root elongation [49]. Studies have also drawn a connection between SLs, GSH, and auxin, where applications of exogenous GR24 increases levels of GSH and causes an inhibitory effect to auxin activity in roots, leading to physiological reductions in lateral root formation. Similar effects were observed in GSH synthesis knockout mutant, suggesting a strong connection between SLs and GSH levels in roots [50].

## 8. Conclusions

By outlining many of its interactions within plant physiology, it is clear that GSNOR has a wide variety of pleiotropic effects in response to its changing environment. At the root level, the architecture of growth is often dependent on the nutrients made available to it by the environment. GSNOR has a role in root structural development, evident through its ability to indirectly mediate nitrate uptake. GSNOR activity may also have a natural system of down regulation in response to oxidative environments, creating a pre-induced antioxidant system. Conversely, phytohormones like auxin production rely on GSNOR activity for growth and transport. Additionally, SLs are an emerging area of research whose role can be further elucidated by studies relating to their relationship with GSNOR. Current studies lack ways of reliably monitoring GSNOR activity. Previous protocols have used NADPH concentration decrease as an indicator for GSNOR activity. This method has limitations in vitro and does not offer a mode of analysis in living tissues where GSNORs pleiotropic effects can be observed. The development of a fluorescent probe to assay GSNOR activity in comparison to known inhibitors and knock out plants would allow for the elucidation of novel roles of GSNOR in plant roots.

## Figures and Tables

**Figure 1 antioxidants-09-01206-f001:**
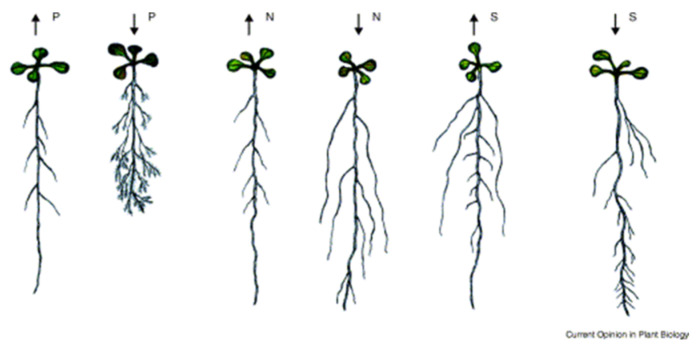
Root development in *Arabidopsis* based of differing macronutrient content [7]. When *Arabidopsis* was grown in growth media containing differing concentrations of P (phosphorus), N (nitrogen), and S (sulfur), there were predictable patterns in root structural development. Reduction in nitrogen levels displayed increases in lateral root formation.

**Figure 2 antioxidants-09-01206-f002:**
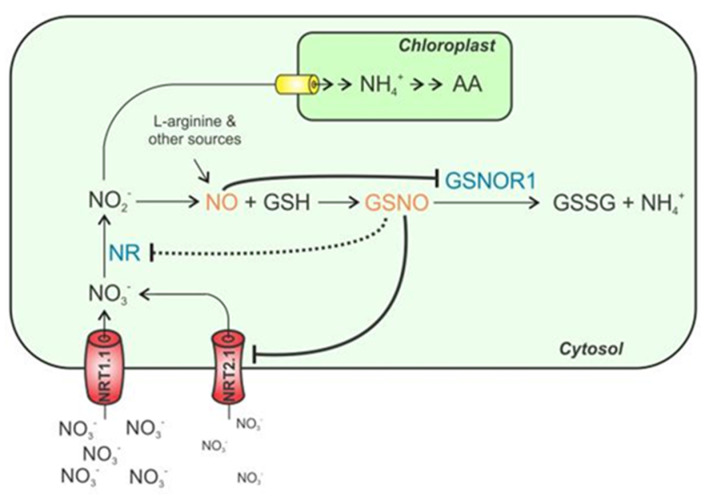
The proposed mechanism for control of the Nitrogen assimilation pathway [30] GSNOR activity is thought to increase NRT2.1 and NR function thereby leading to eventual increases in NO levels, which are ultimately thought to have an inhibitory effect on GSNOR.

**Figure 3 antioxidants-09-01206-f003:**
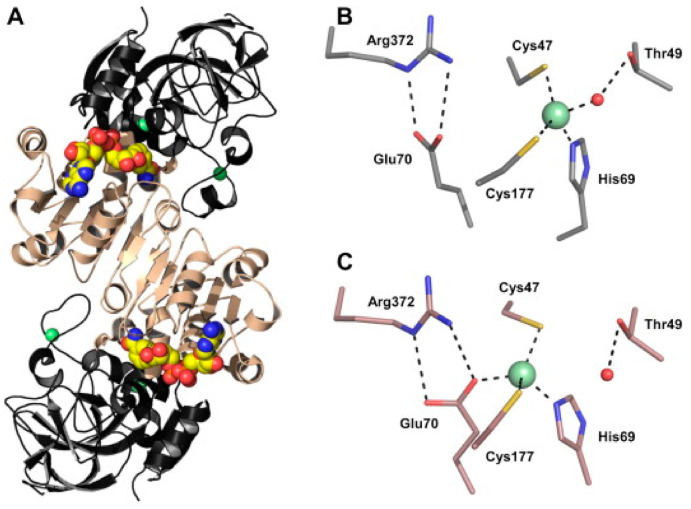
*Solanum lycopersicum* SlGSNOR shown in coordination with NAD^+^ [6] (**A**) the active sites on the homodimer are highlighted, where the coordinated zinc is shown in green (**B**,**C**) a possible point of regulation in the presence of oxidative species.

**Figure 4 antioxidants-09-01206-f004:**
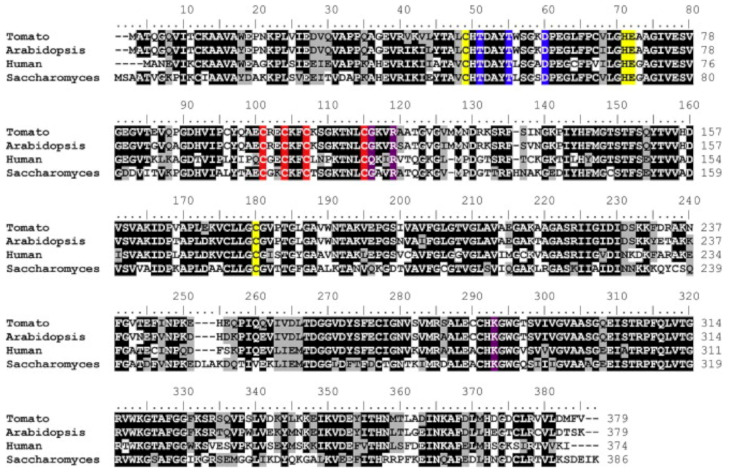
GSNOR amino acid sequence alignment showing conserved residues [6]. There are conserved cysteine residues at cys10, cys271 and cys370; possible S-nitrosation sites.

**Figure 5 antioxidants-09-01206-f005:**
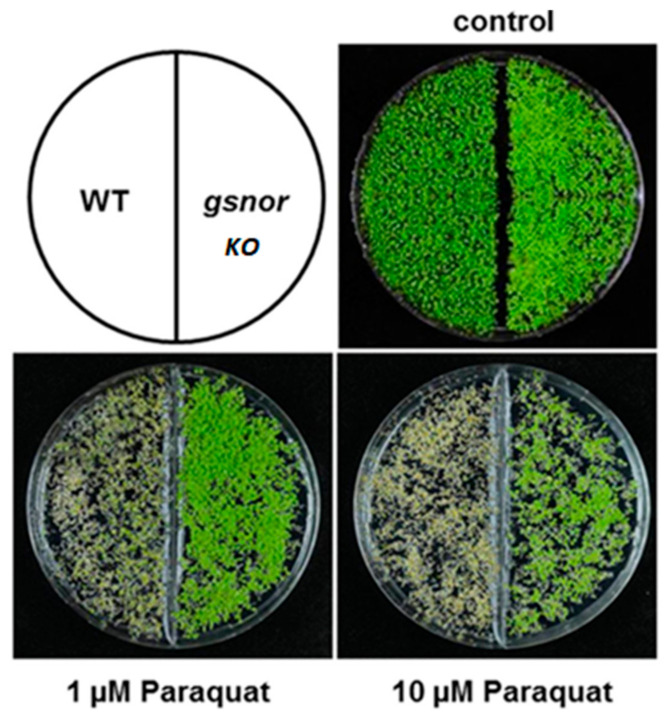
GSNOR knock out has innate resistance to an oxidative environment in Arabidopsis [38]. The upper left image displays the organization of the dish containing sprouted seeds with WT GSNOR on the left and the GSNOR knock out on the right. The bottom images display that with increased concentration of the herbicide, paraquat, which acts through creating a highly oxidative environment there is obvious native resistance in the GSNOR knock out seeds.

**Figure 6 antioxidants-09-01206-f006:**
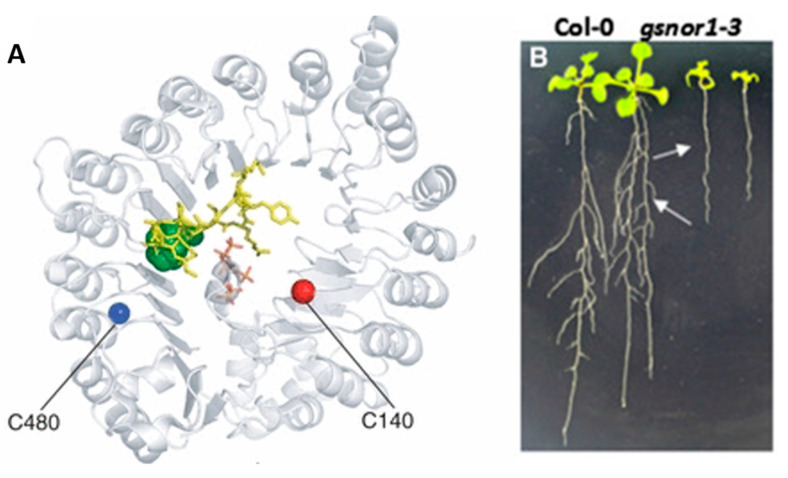
Possible S-nitrosation site on TIR1 not consistent with phenotype of GSNOR 1–3 mutant [1,44]. (**A**) shows S-nitrosylation sites on TIR1 thought to be capable of upregulating auxin production hypothesized to increase growth however (**B**) shows that in application the GSNOR knock out leads to stunted growth, evidence in favor of GSNORs pleiotropic effects.

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
