# Peer review of "The Physiological Implications of *S*-Nitrosoglutathione Reductase (GSNOR) Activity Mediating NO Signalling in Plant Root Structures"

_antioxidants, 2020, doi:10.3390/antiox9121206_

Round 1

Reviewer 1 Report

The manuscrispt by Mutus and Ventimiglia is a short review on the role of S-nitrosoglutathione reductase activity related to NO signalling in plant roots. Although a quick search on the most frequently used research engines shows a lot of articles on the subject this review only has 36 references and as such it fails on showing the relevance of the topic. This is the major problem concerning this manuscript that the authors should consider before publication.

Author Response

Thank-you for your feedback. I appreciate concerns about a lack of references and have attempted to remedy this, however the role of GSNOR in plant roots structures is still a fairly new field of study. There are not many studies that specifically highlight the interactions between GSNOR and the mechanisms of response in roots. 

Reviewer 2 Report

The Manuscript ID: antioxidants-966727- Title: The Physiological Implications of S-nitrosoglutathione Reductase (GSNOR) Activity Mediating NO Signalling in Plant Root Structures by Leslie Ventimiglia and Bulent Mutus.

COMMENTS

This review describes the role of GSNOR activity in the regulation of redox signaling. In my opinion this review is interesting, well written and organized. The language is comprehensive and exhaustive. The figures consistent with the text.

Author Response

Thank-you for the positive feedback 

Reviewer 3 Report

After reading the manuscript several times, I feel the title does not reflect the content. Many similar works have already been published. In my opinion, the new version of the manuscript should focus more on what the authors want to convey regarding the role of this enzyme in root physiology. I would like to ask if permissions are obtained from other editorial offices for the again use of photos in this manuscript? Please also correct the writing of chemical compounds throughout the manuscript , especially the description of the number of atoms in the subscript or additional notations in the superscript. The work requires rethinking and clearly defining what is the main thought and foundation of this work. They are hardly visible in its present form. I am very sorry, but in this form I have to suggest that this version be rejected.

Author Response

Thank-you for your feedback. The intention behind this review is to demonstrate that GSNOR has been identified as a regulatory enzyme in many different stress responses pathways specifically in roots. However, the role of GSNOR in root physiology is still mostly unknown, many studies focus more on NO or auxin in relation to root physiology while more recent studies have begun to to realize the connections with GSNOR. Yes permissions have been obtained and I have corrected the notation of chemical compounds. Hopefully the intention behind this work is apparent in the newest version.

Round 2

Reviewer 1 Report

The authors have addressed the previous comments and made corrections. This version of the manuscript can be published in Antioxidants. 

Reviewer 3 Report

I have read the new version. It has been significantly changed. I still believe that the text could be expanded and further improved. However, in view of the fact that the authors have addressed the previous comments and made corrections, I can accept and recommend this version for publication.